# Unlocking Full Efficiency of Token Filtering in Large Language Model Training

**Di Chai**[1], **Pengbo Li**[2], **Feiyuan Zhang**[2], **Yilun Jin**[2], **Han Tian**[3], **Kaiqiang Xu**[2],
**Binhang Yuan**[2], **Dian Shen**[4], **Junxue Zhang**[3*], **Kai Chen**[2*]
[1]*Shanghai University of Finance and Economics*
[2]*Hong Kong University of Science and Technology*
[3]*University of Science and Technology of China*        [4]*Southeast University*
chaidi@mail.sufe.edu.cn, snowzjx@ustc.edu.cn, kaichen@cse.ust.hk

## Abstract

Token filtering has been proposed to enhance the utility of large language models (LLMs) by eliminating inconsequential tokens during training. While using fewer tokens is expected to reduce computational workloads, existing methods have not yet achieved a real-world efficiency boost. This is primarily due to two factors: (1) existing work has inadequate sparsity for speedup, and (2) token filtering operates within a sparsity range that is non-standard in existing machine learning (ML) libraries and thus cannot be efficiently supported. This paper presents CENTRIFUGE[1], a system that leverages algorithm and system co-design to unleash the full efficiency of token filtering in LLM training. At the algorithm level, CENTRIFUGE filters activations of inconsequential tokens in the attention backward kernel to amplify the sparsity in backward computation. At the system level, CENTRIFUGE proposes an automatic workflow that transforms sparse GEMM into dimension-reduced dense GEMM for optimized efficiency using standard ML libraries. Evaluations on models with various scales—from 1.1B to 40B—demonstrate that CENTRIFUGE reduces backpropagation time by up to 49.9% and end-to-end training time by up to 34.7% when filtering 50% of tokens. Utility assessments indicate that CENTRIFUGE preserves the utility benefits of token filtering and significantly enhances model performance by up to 26.6% compared to standard training. CENTRIFUGE is designed for seamless integration into existing LLM training frameworks, enabling systems already utilizing token filtering to accelerate training with just one line of code.

## 1 Introduction

Training high-quality large language models (LLMs) is notably resource-intensive, requiring substantial investments in both data and computational power. For example, the training process of Llama3-70B occupied approximately 7 million GPU hours over 15 trillion high-quality tokens (Dubey et al., 2024a). *Token filtering* represents an emerging paradigm aimed at enhancing the cost-efficiency of LLM training by systematically discarding less significant tokens early in the training process[2]. This methodology enables the model to concentrate on the most pertinent tokens, resulting in up to 30% absolute improvement in model utility across various tasks (Lin et al., 2024).

While the effectiveness of token filtering in enhancing model utility is well recognized, its potential to improve computational efficiency in training remains largely unexplored. In principle, by significantly reducing the number of tokens processed in the computational pipeline, token filtering should decrease computational demands and expedite training. However, our analysis reveals that combining token filtering with existing LLM training systems can only introduce a marginal 1.2% speedup in end-to-end training time, even when 40% of the tokens are eliminated (§3.1). This limited enhancement in training efficiency constrains the broader advantages of token filtering for large-scale

---

*Junxue Zhang (snowzjx@ustc.edu.cn) and Kai Chen (kaichen@cse.ust.hk) are the corresponding authors.

[1]A centrifuge is a modern system used in chemistry laboratories to efficiently filter different elements.
[2]This paper primarily focuses on backward filtering, as it demonstrates superior performance in enhancing the capabilities of LLMs. Further details can be found in §2.1

LLM training. Therefore, we pose the question: *Can we fully unlock the efficiency of token filtering while simultaneously achieving better utility than conventional training?*

To answer this question, we first investigate the key limitations of the existing token filtering system: (1) Inadequate sparsity and the naive approach of amplifying sparsity cannot work. Existing approaches (Lin et al., 2024) filter tokens at the output layer during loss calculation, which does not create true sparsity. The gradients of these dropped tokens are still computed in the attention backward kernel and propagated to the front layers, leading to dense matrix operations and negating potential efficiency gains. A strawman approach of filtering softmax activations does retain sparsity but cannot work with mainstream memory-efficient attention implementations since softmax outputs are not explicitly stored, and naively filtering activations used for recomputing softmax unintentionally causes interference and harms the gradients. (2) Non-standard sparsity range. Token filtering potentially brings 30∼50% sparsity (Lin et al., 2024), while current sparse matrix multiplication (GEMM) implementations in machine learning (ML) focus on high sparsity range and require >95% sparsity (§3.2) to be effective. Thus, token filtering cannot be supported by existing sparse computations implementations because of mismatching sparsity range. Using these implementations in token filtering actually slows down training efficiency rather than accelerating it (§3.2).

To fully unlock the training efficiency of token filtering and simultaneously achieving better utility than conventional training, we propose CENTRIFUGE. At its core, CENTRIFUGE integrates an algorithm and system co-design:

1. At the algorithm level, CENTRIFUGE carefully analyzes the backward computation and proposes further filtering activations of inconsequential tokens in the attention backward kernel to amplify the sparsity. Our solution is designed to be compatible with mainstream memory-efficient attention implementations (*e.g.*, FlashAttention (Dao et al., 2022)) by separately processing each gradient output to avoid interference. CENTRIFUGE sustains the utility advancements of existing token filtering methods (Lin et al., 2024) and unlocks the potential for efficiency improvement.

2. At the system level, CENTRIFUGE leverages the characteristics of token filtering—specifically, the sparsity of matrices in either columns or rows—to transform sparse GEMM into dimension-reduced dense GEMM, maximizing efficiency on existing machine learning libraries. However, PyTorch's dynamic graph nature complicates global updates to dimensions and variables, as graph variability and node differences prevent static rules for correctness. To overcome this, we design an automatic workflow leveraging the runtime stability (*i.e.*, the graph remains stable during the training) to dynamically identify and update the necessary dimensions and variables before backpropagation.

We implement CENTRIFUGE to be easily integrated into existing training pipelines with minimal code changes. Systems already using backward token filtering only need to add one line of code to achieve efficiency improvement. To better demonstrate the versatility of CENTRIFUGE and its seamless integration with existing training systems, we have broadly tested on CENTRIFUGE on various training scenarios, such as distributed training using tensor parallel (TP) and parameter-efficient fine-tuning using low-rank adapters (LoRA), and CENTRIFUGE has shown consistent efficiency gain.

We comprehensively evaluate CENTRIFUGE in terms of both utility and efficiency using four compact yet powerful models: TinyLlama-1.1B (Zhang et al., 2024), Qwen2.5-1.5B (Yang et al., 2024), Llama3.2-3B, and Llama3.1-8B (Dubey et al., 2024b). The evaluation results indicate that CENTRIFUGE fully preserves the utility benefits of token filtering while increasing model performance by up to 26.6% over standard training methods on nine tasks. With 50% of tokens filtered, CENTRIFUGE reduces backward and end-to-end training cost by up to 49.0% and 31.7%, respectively. Notably, CENTRIFUGE achieves greater efficiency gains in scenarios with high computational demands (*e.g.*, long-context training) and intensive communication (*e.g.*, tensor parallel operations), demonstrating its high practicality in real-world LLM training.

## 2 PRELIMINARY AND RELATED WORK

### 2.1 PRELIMINARY OF TOKEN FILTERING IN LLM TRAINING

Token filtering is a recently proposed technology that has been well recognized by the AI community. The core idea is to identify and filter out tokens that are either noisy or unlikely to contribute meaningfully to the LLM training process, which implicitly improves the quality of training data

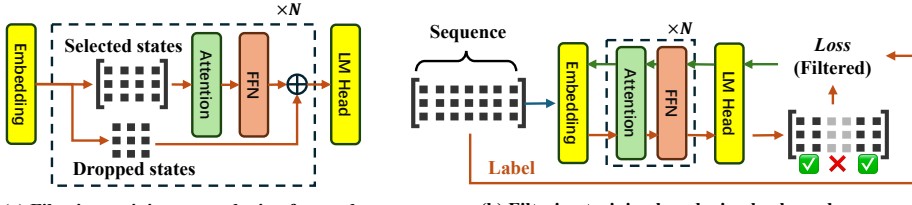

(a) Filtering training states during forward.  (b) Filtering training loss during backward.

Figure 1: An overview of existing token filter studies.

to benefit the model utility. Moreover, by reducing the total number of tokens to be trained, token filtering also brings the opportunity for efficiency improvement. Existing token filtering work can be categorized into two types: *forward token filtering* and *backward token filtering*. As illustrated in Figure 1, forward token filtering techniques remove training tokens during the forward pass, whereas backward token filtering methods eliminate tokens exclusively during the backward pass.

Forward token filtering methods have been extensively studied in previous works (Hou et al., 2022; Zhong et al., 2023; Yao et al., 2022; Ataiefard et al., 2024). However, they typically underperform compared to backward filtering methods due to semantic losses (Zhong et al., 2023; Yao et al., 2022; Lin et al., 2024). As shown in Figure 1, forward token filtering methods filter tokens at each layer of the forward computation, such that each layer of the model only processes partial context. However, this approach has been shown to cause semantic loss and potentially harm model utility (Zhong et al., 2023; Yao et al., 2022). Evaluations in existing forward filtering studies (Hou et al., 2022; Zhong et al., 2023; Yao et al., 2022; Ataiefard et al., 2024) report only similar or lower model utilities and fail to achieve the improvements in utility seen with backward filtering methods (Lin et al., 2024).

In this paper, we focus on backward token filtering due to its impressive advantages in improving model utility. As illustrated in Figure 1, the backward filtering method maintains standard forward computation while performing selective token training in the output layer. Existing studies leverage a reference model to assess the importance of each token. Mathematically, backward token filtering can be formulated as follows (Lin et al., 2024):

$$\mathcal{L}_{filter} = -\frac{1}{N \times k\%} \sum_{i=1}^{N} I_{k\%}(\mathbf{x}_i) \log P_\theta(\mathbf{x}_i | \mathbf{x}_{<i}; \theta) \tag{1}$$

$$I_{k\%}(\mathbf{x}_i) = \begin{cases} 1, \text{ if } \mathbf{x}_i \ \in \ top \ k\% \ of \ (\mathcal{L}_\theta(\mathbf{x}_i) - \mathcal{L}_{ref}(\mathbf{x}_i)) \\ 0, \text{ otherwise} \end{cases} \tag{2}$$

where $\mathcal{L}_\theta$ is the loss of the target model, $\mathcal{L}_{ref}$ is the loss of the reference model, and $\mathcal{L}_{filter}$ is the actual loss to train the target model while keeping $k\%$ of tokens.

## 2.2 RELATED WORK

**Data selection.** Data selection is a pre-processing technique (*i.e.*, applied before training) aimed at improving data quality. Typically, it involves selecting diverse and high-quality training samples (Bai et al., 2024; Wettig et al., 2024; Xie et al., 2023; Fan et al., 2024; Ye et al., 2025; Liu et al., 2025; Wenzek et al., 2020; Thrush et al., 2025). Data selection occurs at the sample level prior to training, which can introduce biases that can negatively impact model utility (Lin et al., 2024). In contrast, CENTRIFUGE functions as a fine-grained data selection method at the token level and, more importantly, selects tokens in a model-adaptive manner by evaluating whether the tokens contributes significantly to improving the model's performance.

**Parameter-efficient training.** Apart from efficient training methods that focus on data sparsity, another branch of research is parameter-efficient training (Ding et al., 2023), which emphasizes sparsity in model parameters. Typical techniques include low-rank adapters (Hu et al., 2022; Dettmers et al., 2023; Karimi Mahabadi et al., 2021; Loeschcke et al., 2024; Valipour et al., 2023), prefix tuning (Li & Liang, 2021), *etc*. CENTRIFUGE and parameter-efficient training differ in their focus on improving training efficiency at the data and parameter levels, respectively. In particular, CENTRIFUGE can work with parameter-efficient training methods to further enhance efficiency. Our experiments demonstrate that CENTRIFUGE is highly generalizable and seamlessly integrates into LoRA training, accelerating backward computation by up to 43.1% when filtering 50% tokens.

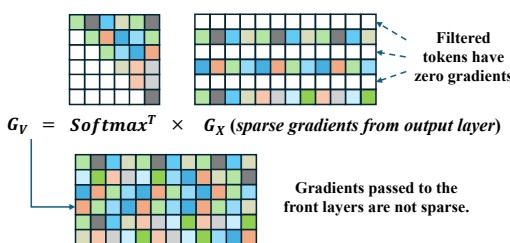
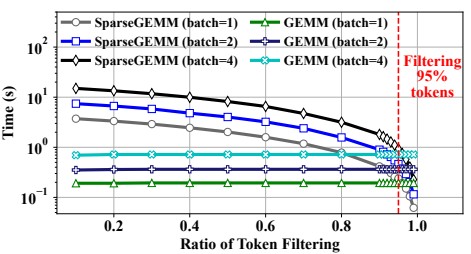

$G_V = Softmax^T \times G_X$ *(sparse gradients from output layer)*

Filtered tokens have zero gradients

Gradients passed to the front layers are not sparse.

(a) Leaving the activation (*e.g.*, *softmax*) of filtered tokens unchanged makes the gradients of attention block being dense matrices, which leads to vanishing gradients.

(b) PyTorch sparse GEMM is effective when sparsity >95% and cannot improve efficiency of token filtering which typically drops 30%∼50% tokens.

Figure 2: Existing token filtering methods exhibit inadequate sparsity, and even when sufficient sparsity is present, sparse GEMM implementations are unable to efficiently support token filtering.

## 3 OBSERVATIONS ABOUT TOKEN FILTERING

We present two key observations that inspire the design of CENTRIFUGE: (1) a deeper understanding of why existing token filtering method has inadequate sparsity and the impracticality of an intuitive strawman approach of amplifying sparsity; (2) profiling results showing the impact of sparsity mismatching between token filtering and sparse computations in ML libraries.

### 3.1 INADEQUATE SPARSITY OF EXISTING TOKEN FILTER

**Inadequate sparsity after token filtering.** Current methods filter the loss of unimportant tokens at the output layer, resulting in sparse gradients. However, they leave all dense activations unchanged. Consequently, after being multiplied by these dense activations, the gradients are no longer sparse once they pass through the first attention block. Therefore, existing backward filtering methods (Lin et al., 2024) exhibit inadequate sparsity, even after filtering the loss at the output layer. Figure 2(a) illustrates the process of computing gradients for $\mathbf{V}$ (*i.e.*, $\mathbf{G_V}$) using sparse gradients while maintaining unchanged activations (*i.e.*, activations of all tokens are retained). After filtering the tokens based on loss, the gradients of the corresponding tokens become zero, as depicted in Figure 2(a). However, because the activations of the filtered tokens remain unchanged, the gradients of $\mathbf{V}$ are no longer sparse. Consequently, the backward computation following the first attention block lacks sparsity, limiting efficiency improvements solely within the output layer.

Following the setting in existing work (Lin et al., 2024), we can estimate the upper bound of efficiency improvement with existing token filtering schemes. Taking TinyLlama, a model with 22 layers and 1.1B parameters, as an example. Filtering 40% tokens will only linearly improve the efficiency on backward propagation of the last layer, while no front layers can be improved. Thus, the backward efficiency can only be improved by 1.8%. Given that backpropagation consumes 66% of the whole training (Narayanan et al., 2021), the end-to-end efficiency improvement is only 1.2%.

**Strawman approach of creating sparsity cannot work.** A simplistic strawman approach to amplify sparsity is that we further filter the activations accordingly. Specifically, the activation of softmax determines the sparsity of $\mathbf{V}$'s gradient and implicitly impacts the gradients of $\mathbf{Q}$ and $\mathbf{K}$ through $\mathbf{G}_A$ (*i.e.*, $\mathbf{G}_A$ is also computed based on softmax). We can filter the activation of softmax, *i.e.*, set the data corresponding to the filtered tokens to zero, to retain the sparsity.

However, this strawman approach cannot work in existing LLM training system that typically uses memory-efficiency attention implementations (*e.g.*, FlashAttention (Dao et al., 2022)). Because: (1) memory-efficient attention does not explicitly store the softmax output, making it impossible to directly filter the softmax activations; (2) if we alternatively filter the $\mathbf{Q}, \mathbf{K}, \mathbf{V}$ activations, which are used for recomputing the softmax and its gradients in backward of memory-efficient attention, an interference will occur between the kernel outputs, since not all gradient outputs impact sparsity and may not require activation filtering. Specifically, the gradients $\partial \mathbf{Q}$ will be unintentionally harmed (§4.1). Based on our experiments, naively using the strawman approach with memory-efficient attention significantly harms model training, causing non-converging training loss (§5.2).

**Observation 1.** Existing token filtering methods exhibit inadequate sparsity because the sparsity fails to propagate from the output layer to the front layers. We require an algorithm that ampli-

fies sparsity and, importantly, is compatible with mainstream memory-efficient attention kernels to achieve true efficiency gains.

## 3.2 MISMATCHING SPARSITY RANGE HINDERS EFFICIENCY GAIN

**Inefficient sparse GEMM.** Existing sparse GEMM implementations are not well-suited for the token filtering training. Although sparse GEMM is a hot research topic and PyTorch has provided a sparse tensor implementation (*i.e.*, `torch.sparse`), the efficiency of existing sparse GEMM is only improved when the data has very high sparsity (*e.g.*, 95%). To demonstrate the problem, we perform experiments on our testbed (details in §5.1). We compare the efficiency of sparse GEMM in PyTorch and regular GEMM in the scenario of token filtering, *i.e.*, the matrix is sparse by row or columns. Figure 2(b) shows the comparison results under different ratios of token filtering and batch sizes. The sparse GEMM is more efficient only when over 95% of all tokens are filtered, which is unrealistic for token filtering. Under the typical filtering rate of 40%, sparse GEMM is even $10\times$ slower than regular GEMM.

**Observation 2.** The valid sparsity range of existing sparse GEMM in ML libraries exceeds 95%, which is misaligned with the 30%~50% sparsity in token filtering. Naively using existing sparse GEMM implementations in token filtering reduces efficiency rather than accelerating it.

## 4 CENTRIFUGE

To solve this problem, we propose CENTRIFUGE, a system leveraging algorithm and system co-design to unleash the full efficiency of token filtering. CENTRIFUGE features two main design points: 1) At the algorithm level, CENTRIFUGE filters the activations in the memory-efficient attention backward kernel to amplify sparsity (§4.1); 2) At the system level, CENTRIFUGE transforms the sparse GEMM to dimension-reduced dense GEMM through automatically updating the backward computation graph to achieve maximum efficiency using existing ML library (§4.2).

## 4.1 AMPLIFYING SPARSITY IN MAINSTREAM ATTENTION IMPLEMENTATION

To properly amplify the sparsity, we need to comprehensively analyze the impact of gradients back-propagation on the desired sparsity. Based on the architecture of decoder-only model, we can categorize the backward process into inter-token and intra-token computations. The inter-token computations happen in self attention block when computing the $softmax(\mathbf{Q}\mathbf{K}^T)\mathbf{V}$, which incorporates computation between tokens. The other computations, including the feed-forward network (FFN), projection of $\mathbf{Q}, \mathbf{K}, \mathbf{V}$, and merge of multi-head attention, are all intra-token computations, which only change the hidden dimensions. In token filtering, our desired sparsity exists in sequence dimension and is only influenced by inter-token computations (*i.e.*, the attention block). Thus we only need to process the attention backward kernel to amplify the sparsity.

Based on observation 1 (§3.1), it is essential to address the issue of inadequate sparsity while ensuring compatibility with mainstream memory-efficient attention implementations to deliver true efficiency improvements. To accomplish this, we analyze the characteristics of the three outputs of memory-efficient attention backward kernel when using token filtering: $\partial\mathbf{Q}$ (query's gradient), $\partial\mathbf{K}$ (key's gradient), and $\partial\mathbf{V}$ (value's gradient).

- $\partial\mathbf{Q}$ have zero values at positions of filtered tokens as long as the input gradients of the attention backward kernel is sparse. Specifically, $\partial\mathbf{Q}$ have the same row sparsity with the gradients of *Attn* based on the attention forward computation $Attn = softmax(\mathbf{Q}\mathbf{K}^T/\sqrt{d})\mathbf{V}$. Thus, $\partial\mathbf{Q}$ is not a factor of causing inadequate sparsity in token filtering.

- $\partial\mathbf{K}$ and $\partial\mathbf{V}$ have non-zero values at positions of filtered tokens since they have the same row sparsity with $\mathbf{Q}$, which is a dense matrix, instead of depending on the input gradients of the attention backward kernel. Thus, we need to filter $\partial\mathbf{K}$ and $\partial\mathbf{V}$ accordingly to retain sparsity.

The above findings indicate that only $\partial\mathbf{K}$ and $\partial\mathbf{V}$ impact sparsity and require filtering to amplify the efficiency. In contrast, $\partial\mathbf{Q}$ remains to be zero for filtered tokens and does not affect sparsity, thus no filtering is needed. Mathematically, we find a conflict: calculating $\partial\mathbf{Q}$ requires the full $\mathbf{K}, \mathbf{V}$ activations, whereas calculating $\partial\mathbf{K}, \partial\mathbf{V}$ requires only part of the $\mathbf{K}, \mathbf{V}$ activations (*i.e.*, the remaining part after filtering). Strawman approach fails because they cannot satisfy these requirements simultaneously. Specifically, when combining the strawman approach with memory-efficient attention,

the strawman method needs to filter the activations of $\mathbf{K}, \mathbf{V}$ in advance to obtain efficiency gain. Unfortunately, this conflicts with calculating $\partial \mathbf{Q}$, which requires the full $\mathbf{K}, \mathbf{V}$ activations. Thus, there is an interference between processing $\partial \mathbf{K}, \partial \mathbf{V}$ and $\partial \mathbf{Q}$.

We propose a novel attention backward kernel that separately processes outputs depending on whether using filtered or non-filtered activations to prevent interference, and the detailed computations are demonstrated in Equation (3) and Equation (4). We can divide the computation into two parts: (1) computing $\partial \mathbf{K}$ and $\partial \mathbf{V}$ using activations of non-filtered tokens (Equation (3)); (2) computing $\partial \mathbf{Q}$ using activations from all tokens (*i.e.*, including the filtered tokens) (Equation (4)). Essentially, the new kernel reorganizes the FlashAttention backward workflow to efficiently compute $\partial \mathbf{K}, \partial \mathbf{V}$ while preserving data integrity for $\partial \mathbf{Q}$.

$$\hat{\mathbf{S}} = \hat{\mathbf{Q}}\hat{\mathbf{K}}^T \quad \hat{\mathbf{P}} = exp(\hat{\mathbf{S}} - \mathbf{L\hat{S}E}) \quad \partial\hat{\mathbf{P}} = \partial\hat{\mathbf{O}}\hat{\mathbf{V}}^T \quad \partial\hat{\mathbf{S}} = \hat{\mathbf{P}} \circ (\partial\hat{\mathbf{P}} - \hat{\mathbf{D}})$$
$$\partial\mathbf{K} = \partial\hat{\mathbf{S}}^T\hat{\mathbf{Q}} \quad \partial\mathbf{V} = \hat{\mathbf{P}}^T\partial\hat{\mathbf{O}} \tag{3}$$

$$\check{\mathbf{S}} = \hat{\mathbf{Q}}\check{\mathbf{K}}^T \quad \check{\mathbf{P}} = exp(\check{\mathbf{S}} - \mathbf{L\hat{S}E}) \quad \partial\check{\mathbf{P}} = \partial\hat{\mathbf{O}}\check{\mathbf{V}}^T \quad \partial\check{\mathbf{S}} = \check{\mathbf{P}} \circ (\partial\check{\mathbf{P}} - \hat{\mathbf{D}})$$
$$\partial\mathbf{Q} = \partial\hat{\mathbf{S}}\hat{\mathbf{K}} + \partial\check{\mathbf{S}}\check{\mathbf{K}} \tag{4}$$

Given an activation $\mathbf{A} \in \mathbb{R}^{b \times s \times h}$, the $\hat{\mathbf{A}} \in \mathbb{R}^{b \times s_1 \times h}$ and $\check{\mathbf{A}} \in \mathbb{R}^{b \times s_2 \times h}$ represent the activations of remaining and filtered tokens, where $b$ is batch size, $s = s_1 + s_2$ is sequence length, and $h$ is the hidden dimensions. We use the same notations with FlashAttention backward algorithm (Dao et al., 2022), where $\mathbf{D}$ is the row sum of $\partial\mathbf{O}$, $\mathbf{LSE}$ is the logsumexp. The kernel output $\partial\mathbf{Q}, \partial\mathbf{K}, \partial\mathbf{V}$ have the same shape of $\mathbb{R}^{b \times s_1 \times h}$, where $s_1$ is the number of remaining tokens. The proposed new attention backward computation is fully compatible with existing memory-efficient attentions, amplifies the sparsity in attention backward kernel, and thereby enables the sparsity propagate from the output layer to all front layers.

## 4.2 DIMENSION-REDUCED DENSE GEMM

While the sparsity is properly amplified from algorithm level, another problem at system level is that existing sparse GEMM implementations cannot effectively support token filtering. Our experiments show that existing sparse GEMM is effective only when the data is highly sparse (*e.g.*, filtering >95% tokens) (§3.2). However, the typical token filtering ratio is 30% ∼ 50% under which existing sparse GEMM implementations even have worse efficiency than dense GEMM.

To address this problem, we propose transforming sparse GEMM into dimension-reduced dense GEMM by leveraging the characteristics of sparsity in token filtering scenarios. We first carefully analyze the characteristics of sparse computation in the backward process. Figure 3 shows the generalized backward process in the computation graph of existing machine learning libraries (*e.g.*, PyTorch), where $\mathbf{G} \in \mathbb{R}^{b \times s \times h_1}$ is the gradients matrix, $b$ is the batch size, $s$ is the sequence length, $h_1$ is the hidden size, $\mathbf{W} \in \mathbb{R}^{h_1 \times h_2}$ is the parameter matrix, and $\mathbf{X} \in \mathbb{R}^{b \times s \times h_2}$ is the input matrix. The gradients of the filtered tokens are set to zero (*i.e.*, $\mathbf{G}$ is row-wise sparse).

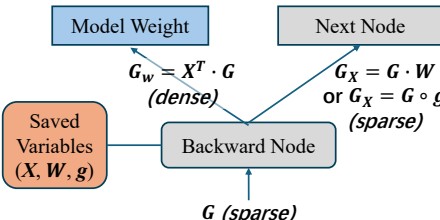

Figure 3: The generalized backward computation in existing ML libraries (*e.g.*, PyTorch). Each node in the computation graph necessarily has the input gradients and output for the next nodes. The model weights and saved variables could be optional and vary in different nodes.

We can categorize the sparse GEMM into two types: 1) Gradients for next nodes: $\mathbf{G}_{sparse} \cdot \mathbf{W}$, $\mathbf{G}_{sparse} \odot \mathbf{g}$, which are sparse and passed to the next nodes. $\mathbf{g}$ is a scaler or vector that performs element-wise computation on the gradients; 2) And the gradients of model parameters: $\mathbf{G}_{spase}^T \cdot \mathbf{X}$,

which will be dense and updated to the model weights. Based on these analyses, we have the following findings:

- The gradients passed to the next nodes inherit the sparsity of the input gradients and the sparsity follows the same pattern as the input gradients (*i.e.*, the gradients of filtered tokens are zeros). Thus if we shrink the sequence dimension at the initial gradients (*i.e.*, removing the zeros), all the afterward gradients will be automatically reduced.

- The sequence dimension vanishes in the gradients of model parameters, which is reasonable since the parameters are independent of the sequence length, and we can directly shrink the sequence dimension to reduce the computational cost.

Instead of directly optimizing the sparse matrix computations, we leverage the above observations and propose to globally reduce the sequence dimension of the gradients and the saved variables in the backward computation graph to accelerate the computation. By shrinking the sequence dimension, we can transform the sparse GEMM to dimension-reduced dense computations with optimized performance. Compared with directly optimizing the sparse GEMM, transforming to dense GEMM is more effective since the dense GEMM has been well optimized in existing ML libraries.

However, implementing the dimension-reduced GEMM is non-trivial due to the dynamics of the computation graph's dynamic structure and node usage, which vary across implementations and inputs, making static updating rules impractical. To address this issue, we propose an automatic workflow to amend the computational graph. The key insight is that, even though the graph is highly dynamic, it still can be deterministic when using the same implementation and inputs. Particularly, the model implementation and inputs remain the same during the whole training (*i.e.*, runtime stability). Thus, we can mimic the input and traverse the graph using the same implementation to dynamically determine the node information. The automatic workflow contains two steps: 1) generating the skeleton code for processing each type of nodes and their attributes; 2) leveraging special markers (*e.g.*, prime numbers) to identify the target updating dimensions and dynamically generating detailed node processing rules. Due to the space limitation, we put the detailed design of automatic graph updating workflow in appendix B.

### 4.3 OVERALL SYSTEM IMPLEMENTATION

CENTRIFUGE improves the efficiency of token filtering through an algorithm and system co-design: a new attention backward kernel to amplify the sparsity from algorithm level, and transforming sparse GEMM to dense GEMM to optimize efficiency from system level. We implement the new attention kernel through leveraging FlashAttention on filtered $\mathbf{Q}, \mathbf{K}, \mathbf{V}$ activations and further computing the remaining $\partial \mathbf{Q}$ using another cuDNN memory-efficient attention kernel that supports attention bias. For non-attention kernels, we directly remove the activations based on the system design. Eventually, for systems that already utilize token filtering, only one line of code `centrifuge.ops.backward_filter(loss, filter_mask)` is needed to get the full efficiency of token filtering. Due to space limitation, we put the implementation details in appendix C.

## 5 EVALUATION

In this section, we comprehensively evaluate CENTRIFUGE in terms of both utility and efficiency. The results demonstrate that CENTRIFUGE fully preserves the utility benefits of token filtering and can improve model performance by up to 26.6% compared to regular training. When filtering 50% of tokens, CENTRIFUGE reduces backward and end-to-end training costs by 40.0%~49.9% and 17.9%~34.7%, respectively.

### 5.1 EXPERIMENTAL SETUP

**Testbed setup.** We evaluate CENTRIFUGE on Ubuntu servers, each equipped with 8 NVIDIA RTX 3090 GPUs (24GB), 40 CPU cores, 256GB of memory, PyTorch 2.8.0 and CUDA 12.8. We implement gradient accumulation to facilitate large batch training and use mixed precision with BF16 to reduce memory consumption and accelerate training.

**Datasets, models, and tasks.** Our experiments focus on fine-tuning pre-trained foundation models for downstream tasks. We utilize small but powerful LLMs in the experiments: TinyLlama-1.1B (Zhang et al., 2024), Qwen2.5-1.5B, Qwen2.5-7B (Yang et al., 2024), Qwen3-14B, Qwen3-

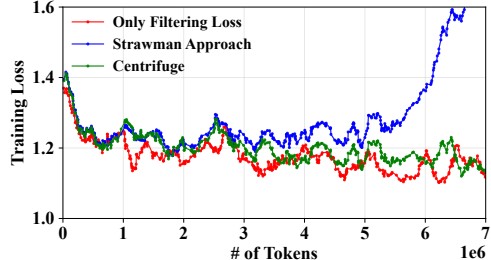 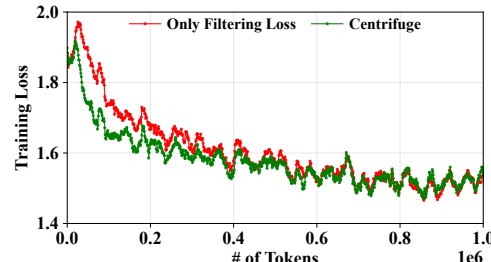

(a) Training loss while using reference model trained on a blend of manually curated math-related tokens.

(b) Training loss while using publicly available model from Lin et al. (2024) as reference.

Figure 4: The training convergence on open-web-math dataset while using different reference models. CENTRIFUGE maintains the utility of token filtering method while the strawman approach fails to converge due to the unintentionally harmed $\partial \mathbf{Q}$.

| Method | Training Dataset | Evaluation Tasks (using TinyLlama-1.1B as foundation model) | | | | | | | | | |
|---|---|---|---|---|---|---|---|---|---|---|---|
| | | GSM8K | MATH | SVAMP | ASDiv | MAWPS | TAB | MQA | MMLU | SAT | Average |
| No Finetuning | NA | 2.3 | 2.4 | 9.9 | 18.1 | 20.2 | 8.8 | 22.1 | 17.9 | 21.9 | 13.7 |
| Regular Finetuning | OWM (Full) | 3.6 | 4.2 | 19.1 | 31.5 | 36.2 | 14.7 | 10.3 | 21.7 | 18.8 | 17.8 |
| CENTRIFUGE (Ours) | OWM (Filter 50%) | 11.8 | 6.4 | 35.5 | 47.4 | 62.8 | 22.4 | 15.3 | 18.7 | 25.0 | 27.3 |

Table 1: Model performance on different tasks. Compared with regular training, CENTRIFUGE significantly improves the model performance by up to 26.6% on single task and 9.5% on average.

32B Team (2025), Llama3.2-3B, Llama3.1-8B (Dubey et al., 2024b), and ALIA-40B Gonzalez-Agirre et al. (2025). Following previous work (Lin et al., 2024), we first train a reference model on small yet high-quality datasets, and then use the loss from the reference model to filter tokens during the training of the target model. We focus on mathematical reasoning as the main task, employing a blend of synthetic and manually curated math-related tokens (Yu et al., 2024; Yue et al., 2024; Mitra et al., 2024; Amini et al., 2019a; Wang et al., 2024) as high-quality data to train the reference model. For large-scale datasets to train the target model, we use open-web-math (OWM) (Paster et al., 2024). We follow (Lin et al., 2024) and use the same architecture for both the reference and target models. To evaluate the utility of the models, we use the following tasks: GSM8K (Cobbe et al., 2021), MATH (Lightman et al., 2024), SVAMP (Patel et al., 2021), ASDiv (Miao et al., 2020), MAWPS (Koncel-Kedziorski et al., 2016), TabMWP (Lu et al., 2023), MathQA (Amini et al., 2019b), SATMath (McAleer, 2023), and MMLU (Hendrycks et al., 2021).

**Baselines and hyperparameters.** We compare CENTRIFUGE with the following baselines: (1) regular training, which involves training the model without token filtering; and (2) token filtering that only filter loss (Lin et al., 2024), *i.e.*, the backward filtering that only perform filtering during loss computation. In convergence evaluation, we aggregate different samples into a context length of 2048, set the batch size to one million tokens, and use a learning rate of $5 \times 10^{-5}$ with cosine decay. In system efficiency evaluation, we report throughput or time consumption averaged on ten iterations after two iterations of warming up.

## 5.2 CONVERGENCE EVALUATION

Figure 4 illustrates the convergence comparison between token filtering that only processes loss, the strawman approach, and CENTRIFUGE on the open-web-math dataset while using different reference models. The results demonstrate that CENTRIFUGE maintains the same utility with token filtering that only filters the loss. Our training system leverages mainstream memory-efficient attention implementation and the strawman approach has failed to converge since it unintentionally harms the **Q**'s gradients during the backward computation as we have discussed in §4.1.

To further demonstrate that CENTRIFUGE achieves better utility than regular training, we compare the performance on math-related tasks and the results are presented in Table 1. The results demonstrate that CENTRIFUGE improves model performance by 9.5% on average. For single tasks, the performance of CENTRIFUGE surpasses that of regular training by up to 26.6% (MAWPS task).

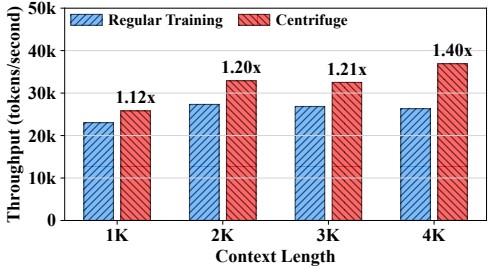 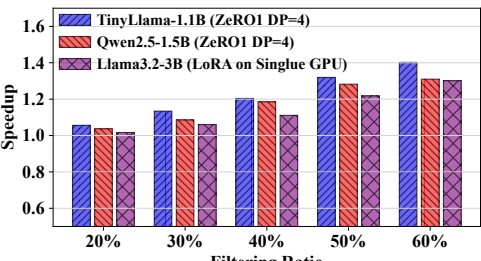

(a) Throughput of regular training and CENTRIFUGE on TinyLlama model with different context lengths. CENTRIFUGE shows superior efficiency, with its efficiency advantage growing at longer contexts.

(b) The end-to-end speedup of CENTRIFUGE compared with regular training on three models with different filtering ratios. The speedup of CENTRIFUGE linearly increases with the filtering ratio.

Figure 5: Efficiency evaluation on different context length and filtering ratio.

| Model | Training Method | Time Consumption of Processing 1 Million tokens (seconds) | | | |
|---|---|---|---|---|---|
| | | Forward | Filter Operator | Backward | Total |
| TinyLlama | Regular Training (DP=4) | 24.75 | / | 53.63 | 79.63 |
| (1.1B, 4K) | CENTRIFUGE | 24.60 | 2.98 | 32.17 ($\downarrow$ 40.0%) | 60.36 ($\downarrow$ 24.2%) |
| Qwen2.5 | Regular Training (DP=4) | 16.03 | / | 35.97 | 52.86 |
| (1.5B, 2K) | CENTRIFUGE | 15.40 | 3.33 | 21.26 ($\downarrow$ 40.9%) | 41.22 ($\downarrow$ 22.0%) |
| Llama3.2 | Regular Training (LoRA) | 34.27 | / | 42.13 | 76.80 |
| (3B, 2K) | CENTRIFUGE | 34.17 | 4.23 | 23.96 ($\downarrow$ 43.1%) | 63.08 ($\downarrow$ 17.9%) |
| Qwen2.5 | Regular Training (TP=4) | 362.54 | / | 811.26 | 1173.81 |
| (7B, 2K) | CENTRIFUGE | 378.61 | 12.84 | 442.25 ($\downarrow$ 45.5%) | 833.72 ($\downarrow$ 28.9%) |
| Llama3.1 | Regular Training (TP=8) | 519.50 | / | 1038.63 | 1558.14 |
| (8B, 2K) | CENTRIFUGE | 520.27 | 14.35 | 529.39 ($\downarrow$ 49.0%) | 1064.02 ($\downarrow$ 31.7%) |
| Qwen3 | Regular Training (TP=8) | 810.35 | / | 1953.37 | 2764.03 |
| (14B, 2K) | CENTRIFUGE | 809.79 | 16.94 | 977.27 ($\downarrow$ 49.9%) | 1804.22 ($\downarrow$ 34.7%) |
| Qwen3 | Regular Training (TP=8) | 95.41 | / | 191.7 | 287.16 |
| (32B, 4K) | CENTRIFUGE | 95.42 | 9.83 | 117.49 ($\downarrow$ 38.7%) | 222.81 ($\downarrow$ 22.4%) |
| ALIA | Regular Training (TP=8) | 116.48 | / | 234.949 | 351.52 |
| (40B, 4K) | CENTRIFUGE | 116.89 | 14.54 | 133.04 ($\downarrow$ 43.4%) | 264.58 ($\downarrow$ 24.7%) |

Table 2: The detailed time consumption on four models using different training methods, including distributed training using DP and TP, and parameter-efficient training using LoRA on single GPU. When filtering 50% tokens, CENTRIFUGE reduces backward time and end-to-end training time by 40.0%∼49.9% and 17.9%∼34.7%, respectively. The results for Qwen3-32B and ALIA-40B are obtained on a testbed comprising eight H20-96GB GPUs interconnected with NVLink, a 96-core CPU, and 1.2TB of memory.

## 5.3 SYSTEM EFFICIENCY EVALUATION

While evaluating the efficiency of CENTRIFUGE, we divide the training process into three stages: forward, filtering operator (*i.e.*, updating the graph), and backward. Table 2 presents a detailed time comparison for these three stages under different scales of models and training methods. In distributed training using ZeRO1 (Rajbhandari et al., 2020) data parallel (DP) and tensor parallel (Narayanan et al., 2021) (TP), CENTRIFUGE significantly improves efficiency of backward computation by 40%∼49.9% and end-to-end training by 22%∼34.7%. Particularly, CENTRIFUGE achieves more speedup in TP since the communication size in TP is also linearly decreased after activation filtering. Thus, CENTRIFUGE is extremely useful in distributed training not only for decreasing the computational cost but also reducing the communication size. Similarly, CENTRIFUGE can also reduce the communication size of pipeline parallel (PP) and mixture-of-expert (MoE) parallel, which is extensively discussed in appendix D. We also evaluate CENTRIFUGE on parameter-efficient training using low-rank adapter (Hu et al., 2022) (LoRA) on a single GPU. The results show that CENTRIFUGE also significantly improves the efficiency by 43.1% in backward computation and 17.9% in end-to-end LoRA training, while setting rank size to 64 and finetuning the attention modular (*i.e.*, projections of $\mathbf{q}$, $\mathbf{k}$, $\mathbf{v}$ and $\mathbf{o}$).

**Overhead of the filtering operator.** We report the overhead of updating the graph in Table 2. Our current implementation updates the graph layer by layer. Consequently, the associated computational cost increases with the model size, which explains why the larger models in Table 2 tend to exhibit higher overhead than the smaller models. However, our experimental results indicate that the cost of the filtering process is substantially lower than the reduction in training time, resulting in a clear improvement in end-to-end efficiency. In particular, the overhead of updating the graph can be further reduced by overlapping the filtering process with computation using two micro-batches, i.e., while one updates the graph, the other executes the forward computation. Additionally, the offline preparation cost of our system is a single forward pass, for getting the overall graph structure, which is minimal compared to the entire training process.

**Impact of the context length.** The context length is a critical factor influencing the training efficiency of LLMs, as the training complexity increases quadratically with the context length. We evaluate the efficiency of CENTRIFUGE using different context lengths on the TinyLlama model. Figure 5(a) illustrates the throughput of regular training and CENTRIFUGE across various context lengths ranging from 1K to 4K. The throughput of CENTRIFUGE consistently exceeds that of regular training, with the efficiency improvement becoming more pronounced as the context length increases. At a context length of 4K, CENTRIFUGE achieves a $1.40\times$ higher throughput. These results demonstrate that CENTRIFUGE can effectively enhance efficiency in computationally intensive scenarios, such as training of long context and large models.

**Impact of the filtering ratio.** To further investigate the efficiency improvement of CENTRIFUGE, we evaluate the end-to-end speedup compared to regular training under different filtering ratios, with the results shown in Figure 5(b). The speedup of CENTRIFUGE increases linearly with the filtering ratio, demonstrating the effectiveness of CENTRIFUGE's system design and indicating potential performance gains in scenarios with higher filtering ratios (*e.g.*, long-context training).

## 6 DISCUSSION

**CENTRIFUGE with more models.** The core designs of CENTRIFUGE are generalizable across diverse model architectures and training frameworks. In the implementation, we adopt an approach that minimizes dependence on specific models and frameworks—specifically, by directly updating nodes in the computation graph. When different models and frameworks are implemented using the same backend (*e.g.*, PyTorch), the underlying computation graph nodes are consistent. Combined with our design for autonomously processing graph nodes, which is detailed demonstrated in Section B, our system can seamlessly support models and frameworks within the same backend.

**CENTRIFUGE with MoE.** We provide detailed analysis in Section D that CENTRIFUGE can benefit different parallelism strategies (*e.g.*, PP, SP, and MoE) by reducing the communication size. We would like to discuss more on MoE since it inherently faces load balancing challenges and integrating CENTRIFUGE with MoE could lead to an interesting research problem. While token filtering would improve overall efficiency, its impact on load balancing remains unclear, as the number of filtered tokens may vary across experts. Therefore, effectively addressing the load balancing issue may be essential to maximize the efficiency gains achieved through token filtering. We identify this as a valuable direction in the future.

**CENTRIFUGE with forward filtering.** Regarding forward filtering, we consider it more suitable for long-sequence training. Existing studies have demonstrated that forward filtering can compromise utility since the context for modeling each token is reduced (Zhong et al., 2023; Yao et al., 2022; Lin et al., 2024), and this problem could be more severe with short sequences. However, for longer sequences (*e.g.*, 128K), recent studies (DeepSeek-AI, 2025) have demonstrated impressive performance in context compression during training. Therefore, integrating long-sequence forward filtering solutions with CENTRIFUGE represents a promising direction for improving the efficiency of long-sequence training in the future.

## 7 CONCLUSION

In this paper, we propose CENTRIFUGE, a system that unlocks the full efficiency of token filtering in LLM training. CENTRIFUGE maintains sparsity by further filtering the activations and transforms sparse GEMM into dense GEMM to optimize efficiency in existing ML libraries. Extensive evaluations demonstrate that CENTRIFUGE effectively achieves its design targets.

ACKNOWLEDGMENTS

We thank the anonymous reviewers for their valuable feedback and suggestions. This work is supported by the Fundamental Research Funds for the Central Universities, NSFC 62402407, Hong Kong RGC TRS T41-603/20R, and Turing AI Computing Cloud (TACC) Xu et al. (2025). Junxue Zhang and Kai Chen are the corresponding authors.

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

# A  THE USE OF LLMS IN WRITING

We used LLM, namely DEEPSEEK-R1, to polish the writing of this manuscript. No other generative AI functionality is used in the writing of this submission.

# B  IMPLEMENTATION DETAIL OF DIMENSION-REDUCED GEMM

## B.1  DYNAMIC GRAPH COMPLICATES THE TRANSFORMATION

Based on observations from backward computations, sparse matrix operations can be reformulated into dimension-reduced dense computations. To implement this proposed approach, it is crucial to first understand the functionality of existing automatic differentiation (autograd) libraries. For example, PyTorch, one of the most widely used frameworks, employs a dynamic computational graph that is constructed incrementally as operations are executed. Other machine learning libraries (*e.g.*, TensorFlow (Abadi et al., 2016)) and systems (*e.g.*, MegatronLM (Narayanan et al., 2021) and DeepSpeed (Rasley et al., 2020)) also use a similar graph-based approach or are mostly built on PyTorch. The backward graph is built during forward propagation and is utilized only once to compute gradients (*i.e.*, discarded after the backward pass in each iteration).

Backward token filtering occurs after the forward computation, at which point the computation graph has already been constructed. Therefore, we need to update the computation graph node by node (*e.g.*, updating the sizes and variables) prior to performing the backward computation. However, modifying the computational graph poses significant challenges due to the following reasons:

- Dynamic graph structure. The computational graph is dynamically built. Different implementations of the same algorithm can have significantly different backward computation graph. Even the input and output can impact the graph, *e.g.*, FlashAttention (Dao et al., 2022) only accept model weights in 16-bits and naive attention implementation is the only choice if we need to explicitly output the attention values. The dynamic graph structure makes it impractical to design static updating rules based on the model (*e.g.*, designing static rules for updating self-attention and FFN layers).

- Dynamic usage of the graph nodes. The same type of node can have different inputs and outputs in different models or even in the same model. For example, the same multiplication node has different outputs when multiplying with scalar, vector, or matrix. The dynamic usage of the graph nodes makes it impractical to use static updating rules based on the node types.

- Numerous types of nodes. Different models typically have different computations and thus use different type of nodes. Different nodes require different updating logic. PyTorch, for instance, has over 300 node types, significantly increasing the complexity of system implementation.

In summary, effectively accelerating the token filtering requires us to transform the sparse GEMM to dimension-reduced dense GEMM that requires updating the computation graph, which is challenging due to the dynamic of graph structure, the dynamic usage of nodes, and the numerous types of nodes.

## B.2  AUTOMATIC GRAPH UPDATING WORKFLOW

To address this issue, we propose an automatic workflow to amend the computational graph. The key insight is that, even though the graph is highly dynamic, it still can be deterministic when using the same implementation and inputs. Particularly, the model implementation and inputs remain the same during the whole training (*i.e.*, runtime stability). Thus, we can mimic the input and traverse the graph using the same implementation to dynamically determine the node information. The automatic workflow contains two steps: 1) generating the skeleton code for processing each type of nodes and their attributes; 2) leveraging special markers to generate detailed node processing rules.

Specifically, we first perform a coarse-grained graph traversing using synthetic data (*i.e.*, mimicking the actual inputs) to collect all the node types and find the target attributes (*e.g.*, sizes and variables) that need to be updated. Table 3 shows examples of the node attributes that need to be amended. We select the list of target attributes and corresponding data types based on Torchgen, which could be easily updated if more attributes needs to be processed. We generate skeleton codes for processing attributes of all the nodes and the implementation detail is presented in §C.1.

| Node Attributes | Data Type | Operations |
|---|---|---|
| InputMetadata | Int Array | Update size |
| SavedVariables | Tensor | Reduce Dimensions |
| Matrix Sizes | Int Array | Update size |
| Matrix # of elements | Int | Update value |

Table 3: Examples of the nodes' attributes that need to be updated during amending the graph.

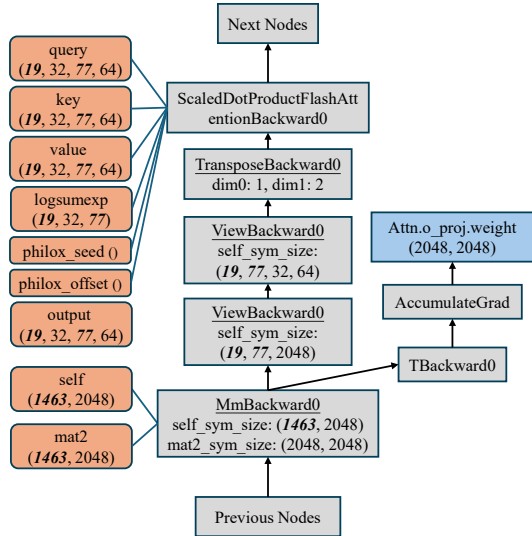

**Special marks: Bsz=*19*, Seq=*77*, Bsz*Seq=*1463***

Figure 6: An example of marking batch size and sequence length with prime numbers. Leveraging the special marks is a simple-but-effective way to precisely find the shrinking dimensions of various nodes in the computational graph.

After obtaining the skeleton code for updating the node attributes, we still need to determine the updating logic. Specifically, we need to determine which dimensions should be reduced and the sizes after the reduction. We only focus on the sequence dimension since we need to filter out unimportant tokens. However, the index of the sequence dimension varies among different nodes and may also be mixed with the batch size. Moreover, the same type of node can have different dimensions at different positions in the same graph, making it impractical to use static updating logic. To address this issue, we design a simple-but-effective method by marking the batch size and sequence length with special numbers to precisely find the shrinking dimensions of various nodes in the computational graph. Figure 6 shows an example of marking the batch size and sequence length with prime numbers. In the skeleton code, we use a greedy algorithm to find the batch size and sequence dimension. Directly using the greedy algorithm can produce wrong results as the batch size or sequence length may be identified with other dimensions (*e.g.*, the sequence length and hidden dimension could be both 2048). The special markers avoid the ambiguity of the dimension and enable the greedy algorithm to precisely find the shrinking dimensions and determine the size after the reduction. The system will cache the output of greedy algorithm for online training (§C.1).

## C  IMPLEMENTATION

We implement CENTRIFUGE in PyTorch, one of the most widely used frameworks, and use its C++ extension[3] to create the backward filtering operator. CENTRIFUGE improves the efficiency of token filtering through two designs: filtering the activations and transforming sparse GEMM to dense GEMM. We implement these two designs in a single operator by directly reducing the sequence dimension, as the filtered activations (*i.e.*, those set to zero) will be subsequently removed in the transformation from sparse GEMM to dense GEMM. Thus, we can directly remove the activations instead of setting them to zero in advance. To address the challenges posed by the dynamic

---

[3]C++ extensions in PyTorch allow users to create custom operators outside the PyTorch backend, providing flexibility and reducing boilerplate code. Once defined, these extensions can be organized into native PyTorch functions for upstream contributions.

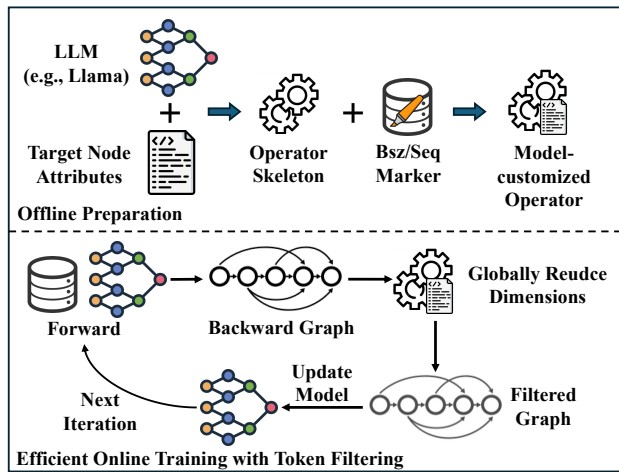

Figure 7: Implementation and usage of CENTRIFUGE.

computation graph and to support various LLM architectures, we implement CENTRIFUGE in two phases: the offline and online stages, as illustrated in Figure 7. In the offline stage, CENTRIFUGE employs an automatic workflow to generate a model-customized operator for updating the graph. In the online training (§C.1), the operator filters the activations and transforms the sparse GEMM into dimension-reduced dense GEMM to accelerate the training process (§C.2). CENTRIFUGE can also be implemented in other frameworks, such as TensorFlow (Abadi et al., 2016), by following similar procedures to update the computation graph during backpropagation.

## C.1 OFFLINE GENERATING MODEL-CUSTOMIZED OPERATOR

Given the model, we run forward and backward computations using synthetic data (*i.e.*, simulating the training samples) to obtain all the node types. Due to runtime stability (*i.e.*, the graph remains stable during training), the node information from synthetic data is identical to that during training. We then parse the node attributes from Torchgen and generate the operator's skeleton code for processing each node. Code 1 shows examples of the generated skeleton code for processing GEMM and FlashAttention nodes. The generated code is a skeleton and cannot be used directly because the operator employs a greedy algorithm to determine the batch size and sequence dimension based on the inputs. However, the actual batch size and sequence length might be the same as other dimensions (*e.g.*, hidden states of 2048), which can mislead the operator into reducing the wrong dimensions.

```
/* $$ start of code generation $$ */
if(fn->name() == "MmBackward0") {
  MmBackward0* op_fn = dynamic_cast<MmBackward0*>(fn);
  auto unpacked_self = op_fn->self_.unpack();
  if(unpacked_self.defined())
  op_fn->self_ = graph_filter->process_variable(unpacked_self, false);
  auto unpacked_mat2 = op_fn->mat2_.unpack();
  if(unpacked_mat2.defined())
  op_fn->mat2_ = graph_filter->process_variable(unpacked_mat2, false);
  graph_filter->process_sizes(op_fn->mat2_sym_sizes);
  graph_filter->process_sizes(op_fn->self_sym_sizes);
}
if(fn->name() == "ScaledDotProductFlashAttentionBackward0") {
  ScaledDotProductFlashAttentionBackward0* op_fn =
    dynamic_cast<ScaledDotProductFlashAttentionBackward0*>(fn);
  auto unpacked_query = op_fn->query_.unpack();
  if(unpacked_query.defined())
  op_fn->query_ = graph_filter->process_variable(unpacked_query, false);
  auto unpacked_key = op_fn->key_.unpack();
  if(unpacked_key.defined())
  op_fn->key_ = graph_filter->process_variable(unpacked_key, false);
  auto unpacked_value = op_fn->value_.unpack();
  if(unpacked_value.defined())
  op_fn->value_ = graph_filter->process_variable(unpacked_value, false);
  auto unpacked_output = op_fn->output_.unpack(op_fn->getptr());
  if(unpacked_output.defined())
  op_fn->output_ = graph_filter->process_variable(unpacked_output, true);
  // ... more attributes omitted
  graph_filter->process_sizes(op_fn->max_q);
```

```
    graph_filter->process_sizes(op_fn->max_k);
}
// ... more nodes
/* $$ end of code generation $$ */
```

Code 1: Generated skeleton code for processing GEMM and FlashAttention nodes. The automatic code generation enables CENTRIFUGE to support various nodes with low implementation costs (*e.g.*, PyTorch has more than 300 types of nodes).

To solve this issue, as demonstrated in §B.2, we compile the generated skeleton code and run the operator using inputs with special markers for both batch size and sequence length. These special markers (*i.e.*, unique from other dimensions) enable the greedy algorithm to precisely identify the correct dimensions for reduction. The operator will save the output of the greedy algorithm and load it during online training, which is guaranteed to be correct due to runtime stability.

After executing the above workflow, we obtain a model-customized operator that contains all the node information and corresponding dimension updating logic for a specific model. We have prepared scripts that allow users to easily execute the workflow and generate operators for their own models. The reset system implementation includes updating the node attributes, *e.g.*, changing the InputMetadata to pass verification and update the saved variables, which is quite straightforward as long as the attributes and reducing dimensions are correctly identified.

### C.2 ONLINE TRAINING USING CENTRIFUGE

```
  import centrifuge
  ...
  for step, batch in enumerate(tokenized_dataset):
    logits = self.model(batch["input_ids"]).logits
-   loss = causal_loss(batch["input_ids"], logits)
+   loss, filter_mask = token_filter_loss(
+       batch["input_ids"], logits,
+       ref_loss=batch["ref_loss"], drop_rate=0.4,
+   )
+   centrifuge.ops.backward_filter(loss, filter_mask)
    loss.backward()
    optimizer.step()
    optimizer.zero_grad()
  ...
```

Code 2: Using CENTRIFUGE in the online training.

Using the operator only requires adding a few lines (*i.e.*, five lines) of code. Code 2 shows an example of using CENTRIFUGE in online training. Starting from regular training, we first need to change the loss computation to a token-filtered loss, *i.e.*, only considering the loss on selected tokens. Then, we call the CENTRIFUGE operator using the loss and filter mask to update the graph. The filter mask is a tensor consisting of zeros and ones to indicate which tokens are filtered. For systems that already utilize token filtering, only one line of code `centrifuge.ops.backward_filter(loss, filter_mask)` is needed to get the full efficiency of token filtering.

### C.3 HOSTING

The system is open-sourced and hosted on GitHub. The link is `https://github.com/Di-Chai/Centrifuge`.

## D CENTRIFUGE REDUCES COMMUNICATION OVERHEADS IN DISTRIBUTED TRAINING

CENTRIFUGE optimizes computational efficiency in backward token filtering by transforming sparse GEMM into dense GEMM, which significantly reduces communication overheads in distributed LLM training. Specifically, CENTRIFUGE updates the entire computation graph, where the gradients and activations are reduced along the sequence dimension. Existing parallelism strategies that enable LLM training on distributed systems typically transfer gradients between different nodes or GPUs during backward computation. Therefore, reducing the sequence dimension of gradients can linearly decrease communication overheads in distributed training. We discuss the advantages of CENTRIFUGE across different parallelism strategies as follows.

- Tensor Parallel (TP) (Narayanan et al., 2021). In TP, the transformer models are typically partitioned along the multiple heads and hidden dimensions. All-reduce of gradients on inputs are required twice (*i.e.*, inputs of FFN and attention block) in each layer's backward computation. CENTRIFUGE can linearly reduce the amount of data transferred in each all-reduce operation. Our evaluation results on training Llama3.1-8B when using TP=8 show that CENTRIFUGE improves end-to-end training efficiency by 31.7% when filtering 50% tokens.

- Sequence Parallel (SP) (Korthikanti et al., 2023). SP is designed to be combined with TP to further reduce the memory usage caused by the redundant activations of dropout and layer normalization. In SP, the sequence dimension is partitioned on multiple devices through all-gather during computing dropout and layer normalization and recovered to partition on hidden dimensions through reduce-scatter. CENTRIFUGE can reduce the communication overhead in the corresponding all-gather and reduce-scatter operations.

- Pipeline Parallel (PP) (Narayanan et al., 2021). The advantages of using CENTRIFUGE in PP is straightforward as PP sequentially transfers the gradients in the graph which are linearly reduced by CENTRIFUGE.

- Mixture-of-Experts (MoE) (Rajbhandari et al., 2022). The integration of CENTRIFUGE in MoE helps in optimizing communication between experts during the backward passes, thereby minimizing the data transferred across devices and enhancing throughput. Specifically, CENTRIFUGE ensures that only gradients of important tokens are routed to the corresponding experts. Concurrently, we also find that integrating our system with MoE leads to a new research problem concerning load balancing, which is discussed in the paper.

