# OpenReview forum: "Unlocking Full Efficiency of Token Filtering in Large Language Model Training"
_ICLR.cc/2026/Conference — ICLR 2026 Poster_

### Official Review · Reviewer_5NLK · 2025-10-19

**Soundness:** 3
**Presentation:** 2
**Contribution:** 3
**Rating:** 6
**Confidence:** 3

**Summary:**

This paper presents CENTRIFUGE, a system that combines algorithm and system-level optimizations to improve the efficiency of token filtering in LLM training. The authors identify two key limitations of existing token filtering approaches: (1) inadequate sparsity propagation beyond the output layer, and (2) mismatch between token filtering sparsity (30-50%) and what sparse GEMM libraries can efficiently handle (>95%). CENTRIFUGE addresses these through filtering activations in attention backward kernels and transforming sparse GEMM into dimension-reduced dense GEMM, achieving up to 49% backward time reduction.

**Strengths:**

1. The paper clearly identifies why existing token filtering methods fail to deliver efficiency gains despite their utility benefits, with strong empirical evidence (Figure 2, Section 3).

2. Testing across four models (1.1B-8B parameters) and multiple training scenarios (DP, TP, LoRA) demonstrates broad applicability. The utility evaluation on 9 tasks shows preserved benefits.

3. The one-line integration for existing systems and consideration of real-world distributed training scenarios (TP, PP, MoE) enhance practical value.

4. The paper is generally well-written with good use of figures to illustrate concepts.

**Weaknesses:**

1. Limited algorithmic novelty. The core contribution (filtering K, V activations while preserving Q) is relatively straightforward once the problem is identified. meanwhile, theoretical analysis of why this approach is optimal and exploration of alternative filtering strategies are missing. The connection to existing memory-efficient attention implementations could also be more rigorously analyzed

2. Insufficient Analysis of Strawman Approach. Section 3.1 claims the strawman approach "cannot work" but provides limited evidence, In particular, figure 4(a) shows non-convergence but lacks quantitative analysis of gradient corruption, ablation studies on different filtering strategies, or detailed explanation of the "interference" mechanism.

**Questions:**

1. How does this interact with other optimizations like FlashAttention-2 or PagedAttention? In particular, how exactly does this integrate with FlashAttention's tiling strategy?

2. What is the overhead of the offline preparation phase for new models?

3. Can you provide theoretical analysis of the gradient approximation quality?

---

> ### Author Response · Authors · 2025-11-24
> **Response to Reviewer 5NLK (1/N)**
>
> We thank the reviewer for the valuable and constructive feedback. Since W1 (concern about novelty), W2 (insufficient strawman Analysis), and Q3 (analysis of gradient approximation) all question the validity and mechanism of our sparsity approach, we address them collectively. Then we address the remaining questions individually.
>
> ### **W1, W2, and Q3. Concern about novelty, insufficient analysis of strawman approach, and analysis of gradient approximation quality.**
>
> **Clarifying novelty and analysis of strawman approach.** We agree with the reviewer that the solution may appear straightforward once the problem is identified. However, one of our core contributions lies in identifying the problem and decoupling the gradient dependencies within complex Memory-Efficient Attention (MEA/FlashAttention) kernels. Mathematically, we identified a conflict: calculating $\partial Q$ requires the full $K, V$ activations, whereas calculating $\partial K, \partial V$ requires only part of the $K, V$ activations (i.e., the remaining part after filtering).
>
> The strawman approach works well when not combining with MEA, since it can directly filter the softmax activations for efficienctly computing $\partial K, \partial V$ and leave all $K,V$ activations for computing $\partial Q$. However, when combining with FlashAttn, strawman approach needs to **filter the $K, V$ activations instead of the softmax**, since softmax is not explicitly stored in MEA kernels.  Unfortunately, this conflicts with calculating $\partial Q$, which requires the full $K, V$ activations. Thus, we claim there is an interference between processing $\partial K, \partial V$ and $\partial Q$ when we use strawman approach with MEA. Our idea is proposing a new backward kernel to reorganize the workflow in MEA kernel to efficiently compute $\partial K, \partial V$ while preserving data integrity for $\partial Q$. Essentially, this design resolves the mathematical deadlock.
>
> In summary, although the idea of filtering KV while keeping Q seems straightforward (but effective), identifying the problem itself and effectively combining the idea with FlashAttn are quite challenging and becomes the indispensable step to unlock the efficiency of token filtering.
>
> **Analysis of the approximation quality**. While deriving strict closed-form error bounds for deep non-linear networks is challenging, we justify the approximation quality through:
>
> 1. Empirical Sensitivity Analysis: Our results indicate that as long as the semantics of the Query side ($\partial Q$) are preserved, LLM training exhibits high tolerance to sparsity in $\partial K$ and $\partial V$. CENTRIFUGE achieves stable convergence, whereas the strawman approach fails to converge. In fact, we conducted extensive parameter searches for the strawman method, yet all attempts failed to converge, which further demonstrates the significance of our design principle of keeping $\partial Q$ unaffected.
> 2. Observations from SOTA (DeepSeek): Our concept of filtering inconsequential $K,V$ activations is highly consistent with the recent DeepSeek Sparse Attention (DSA) technical report [1]. DeepSeek demonstrates that aggressive compression can be applied to the KV cache (where Queries only attend to selected KVs) with minor degradation. This external validation confirms that CENTRIFUGE's approach of filtering KV activations of inconsequential tokens is both reasonable and empirically sound.
>
> [1] DeepSeek-V3.2-Exp: Boosting Long-Context Efficiency with DeepSeek Sparse Attention
>
> ### **Q1. How does this interact with other optimizations like FlashAttention-2 (e.g., tiling strategy) or PagedAttention?**
>
> Thank you for the question. Our implementation is compatible with FlashAttention-2 but does not interact with underlying kernel optimizations, such as the tiling strategy. We reorganize the backward computation workflow of FlashAttention-2 and reuse all kernel optimizations for high performance. PagedAttention is orthogonal to our paper, as it is primarily adopted in LLM inference and designed for optimizing memory usage, whereas our system targets high-performance training and aims to reduce computational and communication costs. Regarding other optimizations for efficient training, e.g., GQA, LoRA, activation recomputation, different parallel strategies, our system can well support. In fact, our evaluations have already included models using GQA, training using LoRA, and different parallelisms.

---

> > ### Author Response · Authors · 2025-11-24
> > **Response to Reviewer 5NLK (2/N)**
> >
> > ### **Q2. What is the overhead of the offline preparation phase for new models?**
> >
> > Thank you for the question. We first show the cost of offline preparation. Then we analyze the compatibility of our system to new models.
> >
> > **Cost of offline preparation.** The offline preparation cost of our system is a single forward pass, for getting the overall graph structure, which is minimal compared to the entire training process.
> >
> > **Compatibility of new models.** Our algorithm and system design are generalizable across diverse model architectures. In the engineering implementation, we adopt an approach that minimizes dependence on specific models or frameworks—specifically, by directly updating nodes in the computation graph. When different models and frameworks are implemented using the same backend (e.g., PyTorch), the underlying computation graph nodes are consistent. Combined with our design for autonomously processing graph nodes, our system can seamlessly support models and frameworks within the same backend. Specifically, we support both built-in and customized PyTorch operations:
> >
> > - For build-in operations in PyTorch (e.g., modulars in `torch.nn`), our implementation provides support.
> > - For customized operations (e.g., inheriting `torch.nn.Module` and customize the `forward` and `backward` functions), our implementations also provides support.
> >
> > Thus, for new models using the PyTorch build-in kernels and customized operations, our system can seamlessly support. The cost of offline preparation is just one time of forward computation for getting the graph structures. In fact, our system will automatically perform this offline preparation instead of manually preparing it. For heavy-customized kernels, e.g., transformer engine in MegatronLM, we have varified that most kernels can be directly supported, but also find some of the kernels may cause an exception. The primary reason is that the kernel is not well implemented, e.g., using saved tensor for caching the tensor size but not use it in the computational garph. For now, our system can completely support PyTorch built-in kernels. We are actively working on these exceptional cases and will open-source a system seamlessly work with most popular training frameworks.
> >
> > In summary, our system can seamlessly support most models and frameworks, with offline preparation being automatically handled. We are also actively testing our system on various models and platforms to extend support for extremely customized kernels.
> >
> > ### **Summary of revision**
> >
> > We extend our sincere gratitude to the reviewer for your valuable and constructive comments, which have enabled us to significantly improve the manuscript. Based on your comments, we have made the following revisions to the paper.
> >
> > Section 4
> >
> > - We add more analysis about why strawman approach cannot work and the interference meachnism.
> > - We analyze the correlation with FlashAttention kernel (i.e., reorganizing the workflow).
> >
> > Section 5
> >
> > - We analyze the overhead of offline preparation.
> >
> > Section 6
> >
> > - We discuss the compatibility with different models and frameworks.

---

### Official Review · Reviewer_VSJV · 2025-10-23

**Soundness:** 2
**Presentation:** 3
**Contribution:** 3
**Rating:** 4
**Confidence:** 3

**Summary:**

Token filtering aims to improve training efficiency by focusing on only the most important tokens. However, current methods only introduce marginal training speedup because they do not fully utilize the token sparsity. This paper introduces CENTRIFUGE, a novel system designed to unlock the full efficiency of token filtering in large language model training. At the algorithm level, CENTRIFUGE proposes filtering activations in the attention backward kernel to amplify the sparsity. At the system level, CENTRIFUGE transforms sparse GEMM into dimension-reduced dense GEMM to maximize the computional efficiency. Experiments show that CENTRIFUGE maintains a similar training loss with the traditional approach of only filtering loss, while reducing training time by 31.7%.

**Strengths:**

1. The paper provides a clear diagnosis of why token filtering has not delivered superior training efficiency. The proposed method is intutive and effectively overcomes current limitations.
2.  Experiments on fine-tuning foundation models show that CENTRIFUGE preserves the benefits of token filtering and reduces end-to-end training time by 31.7%.

**Weaknesses:**

1. Experiments are only conducted on fine-tuning pre-trained foundation models for downstream tasks. However, since the main advantage of the proposed method lies in its training efficiency, it should be validated on computationally intensive pretraining tasks.

**Questions:**

None

---

> ### Author Response · Authors · 2025-11-24
> **Response to Reviewer VSJV**
>
> ### **W1. Insufficient validation on computationally intensive pretraining tasks.**
>
> Thank you for your valuable and constructive comments. The core idea of our system is to improve efficiency by filtering inconsequential tokens during the backpropagation process. Our system benefits both computation and communication, as fewer gradients need to be computed and fewer intermediate results need to be transmitted. Essentially, all backward computations and communications involving the sequence dimension can be accelerated linearly according to the filtering ratio. Therefore, our system is expected to achieve even better performance in computationally and communication-intensive scenarios, such as pre-training tasks. Several empirical results from our experiments validate this expectation:
>
> 1. In Figure 5(a), the efficiency gain increases from $1.12\times$ to $1.4\times$ when context length increases from 1K to 4K, since longer context is more computationally intensive.
> 2. In Table 2, reduction in time consumption under TP ($\downarrow 31.7$%) is more prominent than under DP ($\downarrow$ 22.0%$\sim$24.2%), since TP is more communication-intensive.
>
> We conduct more experiments to demonstrate that our system can efficiently support more computational and communication-intensive tasks, i.e., larger models.
>
> |   Model    |     Training Method     | Forward | Filter Op |           Backward           |             Total             |
> | :--------: | :---------------------: | :-----: | :-------: | :--------------------------: | :---------------------------: |
> | Qwen3-14B  | Regular Training (TP-8) | 810.35  |    NA     |           1953.37            |            2764.03            |
> | Qwen3-14B  |       Our System        | 809.79  |   16.94   | 977.27 ($\downarrow 49.9$%) | 1804.22 ($\downarrow 34.7$%) |
> | Qwen3-32B* | Regular Training (TP-8) |  95.41  |    NA     |            191.7             |            287.16             |
> | Qwen3-32B* |       Our System        |  95.42  |   9.83    | 117.49 ($\downarrow 38.7$%) | 222.81 ($\downarrow 22.4$%)  |
> | ALIA-40B*  | Regular Training (TP-8) | 116.48  |    NA     |           234.949            |            351.52             |
> | ALIA-40B*  |       Our System        | 116.89  |   14.54   | 133.04 ($\downarrow 43.4$%) | 264.583 ($\downarrow 24.7$%) |
>
> (*Experiments of 32B and 40B use a new testbed of 8xH20-96GB interconnected with NVLink, 96-Core CPU, and 1.2TB memory).
>
> Essentially, we follow the pre-training settings used in the above experiments, performing full-parameter training, to validate that our system performs well on computationally intensive pre-training tasks. The results demonstrate that our system improves the end-to-end efficiency by 22.4%$\sim$34.7%.
>
> ### **Summary of revision**
>
> We extend our sincere gratitude to the reviewer for your valuable and constructive comments, which have enabled us to significantly improve the manuscript. Based on your comments, we have made the following revisions to the paper.
>
> Section 5
>
> - We extend efficiency evaluation to larger models (i.e., 14B$\sim$40B) with full-parameter training to address the reviewer's concern about insufficient evaluation on computational intensive pre-training tasks.

---

> > ### Comment · Reviewer_VSJV · 2025-11-27
> >
> > Thank you for the response. I don’t doubt the effectiveness of Centrifuge in terms of training acceleration, which has been sufficiently demonstrated in the paper. However, whether Centrifuge achieves a similar loss during pre-training compared to Only Filtering Loss or standard cross-entropy without filtering is, in my opinion, something that needs to be verified.

---

> > > ### Author Response · Authors · 2025-11-28
> > > **Additional experiments on pre-training task**
> > >
> > > Thank you for your reply and raising your concern regarding loss convergence during pre-training. We conduct additional experiments on continued pre-training with general-domain data, and the results are reported in the following figure link. The experiments are performed using Tinyllama 1.1B model and 1 billion tokens sampled from SlimPajama [1], and using open-source model from [2] as the reference. The results indicate that our system achieves a similar loss during continued pre-training compared to only filtering the loss.
> > >
> > > Anonymous link to the experimental figure: https://anonymous.4open.science/api/repo/paperrebuttal-7E0D/file/convergence_3.pdf
> > >
> > > The newly added general-domain pre-training experiments, together with the math-related training tasks already presented in the paper, collectively demonstrate that our system does not adversely affect model convergence.
> > >
> > > [1] SlimPajama: A 627B token cleaned and deduplicated version of RedPajama
> > >
> > > [2] Rho-1: Not All Tokens Are What You Need

---

> ### Comment · Reviewer_VSJV · 2025-11-28
>
> Thank you for your response. I will raise the score to 6 when possible.

---

> > ### Author Response · Authors · 2025-11-28
> > **Response to Reviewer VSJV**
> >
> > Thank you for your valuable comments and for your willingness to raise the score. We sincerely appreciate the time you have taken to review our work.

---

### Official Review · Reviewer_P85p · 2025-10-26

**Soundness:** 4
**Presentation:** 3
**Contribution:** 4
**Rating:** 8
**Confidence:** 4

**Summary:**

This work tackles the problem of improving training efficiency for LLM via token filtering. Prior work showed token filtering can boost model utility/quality but fails to yield meaningful speedups in practice. The authors identify two key obstacles: 1. Insufficient sparsity propagation: existing token filtering (applied only at the loss layer) produces sparse gradients at the output, but upstream layers still process all tokens, negating potential speed gains. Naively filtering intermediate activations breaks standard attention backpropagation and harms gradient correctness. 2. Unsupported sparsity regime: Token filtering typically drops ~30-50% of tokens which leads to a "moderate" sparsity level at which standard sparse deep learning libraries (optimized for >=95% sparsity) become inefficient or even slower. To address these issues, the paper proposes an integrated solution called CENTRIFUGE, with contributions at both the algorithm and system levels:

* Algorithm level: introduces a modified attention backward pass that filters out activation for unimportant tokens during backprop, amplifying sparsity in the gradient computation. This new backward kernel avoids interfering with the attention softmax by processing each output’s gradient separately, which preserves the correct gradient flow and the model quality gains of token filtering. This design is also compatible with high-performance attention implementations (e.g. FlashAttention)

* System level: develops an automated workflow to convert sparse operations into smaller dense operations that can run efficiently on existing libraries. CENTRIFUGE exploits the structure of token sparsity (i.e. whole rows or columns dropped) to shrink the matrix multiplications, achieving the effect of sparse computation by adjusting tensor dimensions at runtime. The authors devise a robust method to dynamically alter the computation graph (despite PyTorch’s dynamic nature) by detecting stable graph patterns and updating tensor shapes before each backward pass. This allows the "moderate" sparsity to be leveraged for real speedups using standard GPU kernels instead of generic sparse GEMM routines which has high overheads.

Experiments on four LLMs (1.1B to 8B parameters) fine-tuned on a large math reasoning dataset show that this method achieves significant speedup with on-par or improved model performance. For example, filtering 50% tokens cut down backward pass time by up to 49% and end-to-end training time by up to 31.7%. Test performance is improved by up to 26.6% on individual tasks (+9.5% on average) comparing to no-filtering baseline.

**Strengths:**

1. The authors identify a crucial bottleneck (lack of real efficiency gains from token filtering) and provide a novel solution with algorithm and system co-design, effectively addressing both the ML aspect (ensuring gradients remain correct and useful) and the systems aspect (making use of sparsity in existing hardware/software). By further filtering attention activations in the backward pass in a safe way, they increase sparsity where it matters, and by transforming sparse ops to dense via graph manipulation, they avoid the usual pitfalls of sparse computation overhead. Together, these yield a unique solution that had not been explored before in literature on token filtering.
2. The experimental evidence is compelling and comprehensive. The achieved speedups are significant (nearly half the backprop time eliminated, and ~30% overall training time reduction at 50% token drop); and importantly, the approach does not harm model quality but in fact improves it. The experiment coverage is also comprehensive: multiple models (ranging 1B to 8B parameters) and different training regimes (single GPU, multi-GPU with DP and TP, and fine-tuning with LoRA) are all tested. This diversity builds confidence that the method is robust and general.
3. The work has high practical significance. Two facts of CENTRIFUGE lower the barrier to adoption: 1) it can be integrated with only one extra line of code for users already doing token filtering, and 2) it's compatible with common distributed training (DP+TP) and in fact further improve throughput by reducing gradient communication payload (Appendix D).

**Weaknesses:**

1. The experiments, although extensive for the provided settings, are limited to relatively small-scale dense LLMs and a specific domain (mathematical reasoning). It remains an open question how well CENTRIFUGE would scale to much larger LLMs (e.g. 70B+ params) or to training on more diverse, general-domain data. Larger LLM might introduce new bottlenecks or slightly different execution characteristics (communication load, optimizer overheads) that were not encountered at 3–8B scale. It might also be helpful to evaluate how the effect of CENTRIFUGE might change in MoE.
2. Forward pass remains unoptimized: Centrifuge focuses on filtering tokens during backprop and does not skip any computation in the forward pass. This design is intentional to preserve model utility, but it means the maximum possible speedup is inherently limited. It would be valuable for the authors to clarify this limitation in the paper. Currently, the paper implies this in Section 2.1 and via citations, but a more explicit discussion of forward-pass cost might be helpful to set expectations for the reader.
3. The system-level solution, involving dynamic graph rewriting in PyTorch, is quite complex. While the authors do an admirable job automating it, one might worry about maintainability. If underlying libraries or model architectures change, these techniques might need adjustment. A discussion on how robust the implementation is to different model architectures (beyond the ones tested) and framework versions would be useful. This isn’t a flaw in the core idea per se, but acknowledging the complexity and how it’s managed would improve transparency.

**Questions:**

Echoing points listed in Weaknesses section:
1. How well do the authors expect CENTRIFUGE to scale to much larger models (e.g., 70B or 100B parameters) and training on massive corpora?
2. Have the authors considered evaluating CENTRIFUGE on different domains or tasks beyond math reasoning, such as coding?
3. Could some form of forward token filtering be incorporated in future to further boost efficiency, and what might be the impact on utility?

In addition, it would be also interesting to know:
4. In the experiments, a fixed fraction of tokens (e.g. 50%) is filtered each step. Have the authors considered making the filtering threshold adaptive, for instance, filtering more aggressively as training progresses or varying by batch?
5. Can CENTRIFUGE be combined with other training efficiency methods (e.g., MoE layers, gradient checkpointing (I think so), or data selection at the example level)?
6. Considering the practical importance, do the authors plan to open-source CENTRIFUGE or integrate it into common libraries like DeepSpeed/FSDP, Megatron, HF/Lightning trainer?

---

> ### Author Response · Authors · 2025-11-24
> **Response to Reviewer P85p (1/N)**
>
> Thank you for your detailed, insightful, and encouraging feedbacks. We response the weakness and questions one by one.
>
> ### **W1, Q1, and Q2. How do the authors expect the system to perform on larger models, massive corpora and diverse domains?**
>
> Thank you for raising the concerns. We first demonstrate that our system is expected to perform well on larger models and massive corpora. Then, we discuss the expected performance on more diverse domains (e.g., coding). Finally, we identify a new research quesiton inspired by the reviewer's question about combing our system with MoE.
>
> **Expected performance on larger models.** By filtering inconsequential tokens during the backpropagation, our system benefits both computation and communication, as fewer gradients need to be computed and fewer intermediate results need to be transmitted. Thus, our system is expected to scale well on larger models and massive corpora since they are computationally and communication-intensive. We conduct additional experiments using 14B$\sim$40B models to demonstrate this.
>
> |   Model    |     Training Method     | Forward | Filter Op |           Backward           |             Total             |
> | :--------: | :---------------------: | :-----: | :-------: | :--------------------------: | :---------------------------: |
> | Qwen3-14B  | Regular Training (TP-8) | 810.35  |    NA     |           1953.37            |            2764.03            |
> | Qwen3-14B  |       Our System        | 809.79  |   16.94   | 977.27 ($\downarrow 49.9$%) | 1804.22 ($\downarrow 34.7$%) |
> | Qwen3-32B* | Regular Training (TP-8) |  95.41  |    NA     |            191.7             |            287.16             |
> | Qwen3-32B* |       Our System        |  95.42  |   9.83    | 117.49 ($\downarrow 38.7$%) | 222.81 ($\downarrow 22.4$%)  |
> | ALIA-40B*  | Regular Training (TP-8) | 116.48  |    NA     |           234.949            |            351.52             |
> | ALIA-40B*  |       Our System        | 116.89  |   14.54   | 133.04 ($\downarrow 43.4$%) | 264.583 ($\downarrow 24.7$%) |
>
> (*Experiments of 32B and 40B use a new testbed of 8xH20-96GB interconnected with NVLink, 96-Core CPU, and 1.2TB memory).
>
> The results show that our system scale well under different model size from 14B to 40B, and improve the end-to-end training efficiency by 22.4% $\sim$ 34.7%.
>
> **Expected performance on more diverse domains**. Regarding training on more diverse datasets (e.g., coding) and general-domain data, existing studies have demonstrated that the native backward filtering algorithm can effectively support various domains, including mathematical reasoning, coding, tool usage, and general-domain data. Due to the limitted computational resources, we only report convergence on math-related tasks. It is anticipated that our system will also perform effectively in these domains. Nevertheless, we agree with the reviewer that validation across more scenarios is essential. We will report additional experimental results alongside the open-source code in the future.
>
> **How the effect of CENTRIFUGE might change in MoE?** We thank the reviewer for the insightful question. After an in-depth analysis, we find that integrating our system with MoE could lead to an interesting research direction. The key reason is that MoE inherently faces load balancing challenges. While token filtering would improve overall efficiency, its impact on load balancing remains unclear, as the number of filtered tokens may vary across experts. Therefore, effectively addressing the load balancing issue may be essential to maximize the efficiency gains achieved through token filtering. We have identified this as a valuable direction for future work in the paper.

---

> > ### Author Response · Authors · 2025-11-24
> > **Response to Reviewer P85p (2/N)**
> >
> > ### **W2 and Q3. Could forward token filtering be incorporated in future and what might be the impact on utility?**
> >
> > Thank you for raising this insightful question. Regarding forward filtering, we consider it more suitable for long-sequence training. Existing studies have demonstrated that forward filtering can compromise utility since the context for modeling each token is reduced, and this utility loss is particularly significant with short sequences, e.g., 4K during pretraining and finetuning. However, for longer sequences (e.g., 128K), recent studies [1] have demonstrated impressive performance in context compression during training. Therefore, we propose that integrating long-sequence forward filtering solutions with our system represents a promising direction for improving the efficiency of long-sequence training in the future.
> >
> > [1] Native Sparse Attention: Hardware-Aligned and Natively Trainable Sparse Attention
> >
> > ### **W3. Limited transparency in system implementation.**
> >
> > Thank you for raising this concern. We now discuss the compatibility of our system with different models the frameworks to improve transparency.
> >
> > **Support of different models and frameworks**. Our algorithm and system design are generalizable across diverse model architectures and training frameworks. In the engineering implementation, we adopt an approach that minimizes dependence on specific models and frameworks—specifically, by directly updating nodes in the computation graph. When different models and frameworks are implemented using the same backend (e.g., PyTorch), the underlying computation graph nodes are consistent. Combined with our design for autonomously processing graph nodes, our system can seamlessly support models and frameworks within the same backend. Specifically, we support both built-in and customized PyTorch operations:
> >
> > - For build-in operations in PyTorch (e.g., modulars in `torch.nn`), our implementation can support.
> > - For customized operations (e.g., inheriting `torch.nn.Module` and customize the `forward` and `backward` functions), our implementations also can support.
> >
> > Furthermore, we plan to make the following efforts to improve transparency.
> >
> > - We will fully open-source the system and experimental scripts in the future to improve transparency.
> > - We will maintain easy-to-read documentation demonstrating the underlying implementations of our system, guidelines for supporting new kernels, and tools for visualizing the computational graph for debugging.
> >
> > ### **Q4. Have the authors considered making the filtering threshold adaptive?**
> >
> > Thank you for raising this insightful question. Implementing an adaptive threshold is an excellent idea, as different samples may have varying significance to the model, and the model's knowledge evolves over time. Intuitively, adaptive filtering ratios could be adjusted from two perspectives:
> >
> > - Adaptive ratios between different batches, using the excess loss as a metric; for example, a higher excess loss may correspond to a lower filtering ratio, as it could indicate that the current model has not yet learned the sample effectively.
> >
> > - Adaptive ratios across the entire training timeline; for instance, gradually increasing the filtering ratio as training progresses, since the model becomes more knowledgeable over time.
> >
> > We will mark this as a future direction in the revised paper. Thank you again for your insightful comments.
> >
> > ### **Q5. Can CENTRIFUGE be combined with other training efficiency methods?**
> >
> > Thank you for your question. Yes, as discussed in Appendix D, our system can be combined with additional parallelism strategies. Integrating our system with gradient/activation checkpointing is certainly feasible from a system perspective; our solution can reduce the I/O overhead when loading activations or the computational overhead of recomputing them. Combining it with data selection at the example level is also a promising direction, as low-quality samples could be filtered in advance to further improve efficiency. We can also develop more refined metrics for sample selection. For instance, samples that are too challenging for the reference model may not be suitable for training the target model.
> >
> > ### **Q6. Do the authors plan to open-source CENTRIFUGE or integrate it into common libraries?**
> >
> > Thank you for your question. Yes, we will open-source the code and integrate it into DeepSpeed, Megatron-LM, and HuggingFace (e.g., accelerate, peft), among others, to make our system truly helpful for research studies and industrial applications.

---

> > > ### Author Response · Authors · 2025-11-24
> > > **Response to Reviewer P85p (3/N)**
> > >
> > > ### **Summary of revision**
> > >
> > > We extend our sincere gratitude to the reviewer for your valuable and constructive comments, which have enabled us to significantly improve the manuscript. Based on your comments, we have made the following revisions to the paper.
> > >
> > > Section 5
> > >
> > > - We add efficiency evaluation on larger models (i.e., 14B$\sim$40B).
> > >
> > > Section 6
> > >
> > > - We discuss the compatibility of our system with various models and frameworks.
> > >
> > > - We discuss the future direction of combining our system with MoE.
> > > - We discuss the future direction of combining our system with forward filtering.

---

### Official Review · Reviewer_6NpQ · 2025-10-31

**Soundness:** 3
**Presentation:** 2
**Contribution:** 3
**Rating:** 6
**Confidence:** 3

**Summary:**

This paper proposes CENTRIFUGE, a system that unlocks the full efficiency of token filtering in LLM training through algorithm-system co-design. At the algorithm level, it filters activations in the attention backward kernel to amplify sparsity while maintaining compatibility with memory-efficient attention implementations. At the system level, it transforms sparse GEMM into dimension-reduced dense GEMM through automatic graph updating. Experiments on four models (TinyLlama-1.1B to Llama3.1-8B) show up to 49% backward time reduction and 31.7% end-to-end speedup when filtering 50% tokens, while achieving 26.6% utility improvement over standard training.

**Strengths:**

1. The paper clearly identifies why existing token filtering methods fail to achieve efficiency gains (inadequate sparsity propagation and sparsity range mismatch with ML libraries), with solid empirical evidence.
2. The paper evaluates across multiple dimensions: different model sizes, various training scenarios, context lengths, and filtering ratios, demonstrating consistent gains and broad applicability.

**Weaknesses:**

1. All experiments use models ≤8B parameters. Given that efficiency gains are most critical for larger models (70B+), the absence of such experiments is a significant limitation. The 8B model uses TP=8, but modern large-scale training uses more complex parallelism (TP+PP+DP). How CENTRIFUGE scales to 70B+ models with TP=8, PP=4 setups remains unknown.
2. The automatic graph updating approach relies on "runtime stability" and special prime number markers, which may be fragile. The paper doesn't discuss failure cases, what happens when graph structure changes unexpectedly, or computational overhead of the offline preparation phase.

**Questions:**

1. Table 2: Why does the filter operator overhead vary significantly (2.98s for TinyLlama vs 14.35s for Llama3.1)? Is this linear in model size or are there other factors?
2. Why only evaluate Qwen2.5-1.5B but not Qwen2.5-7B, especially when Llama3.1-8B is included?

---

> ### Author Response · Authors · 2025-11-24
> **Response to Reviewer 6NpQ (1/N)**
>
> Thank your for your valuable and constructive comments. We address the weaknesses and questions one by one as follows.
>
> ### **W1. All experiments use models ≤8B parameters. How CENTRIFUGE scales to 70B+ models with TP=8, PP=4 setups remains unknown.**
>
> Thank you for raising this concern. We will first demonstrate that our system is expected to have good performance gain under larger models. Then, we will discuss the compatibility with a broader range of parallelism strategies.
>
> **Expected performance under larger models**. By filtering inconsequential tokens during backpropagation, our system benefits both computation and communication, as fewer gradients need to be computed and fewer intermediate results must be transmitted. Consequently, our system is expected to scale effectively with larger models which are inherently computationally and communication-intensive.
>
> Empirical results from our experiments confirm that the system performs particularly well on computationally and communication-intensive tasks.
>
> - In Figure 5(a), the efficiency gain increases from $1.12\times$ to $1.4\times$ when context length increases from 1K to 4K, since longer context is more computationally intensive.
> - In Table 2, reduction in time consumption under TP ($\downarrow 31.7\%$) is more prominent than under DP ($\downarrow 22.0\%\sim24.2\%$), since TP is more communication-intensive.
>
> We conduct additional experiments to verify the scalability of our approach with larger models.
>
> |   Model    |     Training Method     | Forward | Filter Op |           Backward           |             Total             |
> | :--------: | :---------------------: | :-----: | :-------: | :--------------------------: | :---------------------------: |
> | Qwen3-14B  | Regular Training (TP-8) | 810.35  |    NA     |           1953.37            |            2764.03            |
> | Qwen3-14B  |       Our System        | 809.79  |   16.94   | 977.27 ($\downarrow 49.9$%) | 1804.22 ($\downarrow 34.7$%) |
> | Qwen3-32B* | Regular Training (TP-8) |  95.41  |    NA     |            191.7             |            287.16             |
> | Qwen3-32B* |       Our System        |  95.42  |   9.83    | 117.49 ($\downarrow 38.7$%) | 222.81 ($\downarrow 22.4$%)  |
> | ALIA-40B*  | Regular Training (TP-8) | 116.48  |    NA     |           234.949            |            351.52             |
> | ALIA-40B*  |       Our System        | 116.89  |   14.54   | 133.04 ($\downarrow 43.4$%) | 264.583 ($\downarrow 24.7$%) |
>
> (*Experiments of 32B and 40B models use a new testbed of 8xH20-96GB interconnected with NVLink, 96-Core CPU, and 1.2TB memory).
>
> The results demonstrate that our system scales effectively to larger model sizes, from 14B to 40B parameters, and improves end-to-end training efficiency by 22.4%$\sim$34.7%.
>
> **Compatibility with more parallelism strategies.** Apart from the DP and TP we have evaluated in the paper, other frequently used parallelism strategies, such as pipeline parallelism (PP), sequence parallelism (SP), and mixture-of-experts (MoE), are also expected to yield performance gains with our system. The reasons are as follows:
>
> - **PP** sequentially transfers gradients from the last stage to the first stage. Our system benefits PP in two aspects: (1) the gradients transmitted between stages can be linearly reduced; (2) the computational time within each stage can be reduced. We are actively working on the intergrattion with megatronlm to support PP in the future.
> - **SP** is designed to be combined with TP to further reduce the memory usage resulting from the redundant activations of dropout and layer normalization. In SP, the sequence dimension is partitioned across multiple devices via all-gather operations during the computation of dropout and layer normalization and is subsequently recovered for the partitioning of hidden dimensions through reduce-scatter operations. Our system can reduce the communication overhead in the corresponding all-gather and reduce-scatter operations.
> - The integration of our system with **MoE**​ helps reduce communication when assigning tokens to experts during backward passes. Concurrently, we also find that integrating our system with MoE may lead to a new research problem concerning load balancing, which is a valuable future direction.
>
> **Future plan of supporting more parallelisms** Our current implementation is based on PyTorch kernels and is therefore naturally compatible with libraries such as DeepSpeed, Accelerate, and others that use PyTorch as a backend. In the future, we plan to further integrate our solution into Megatron-LM to simultaneously support PP, SP, etc.

---

> > ### Author Response · Authors · 2025-11-24
> > **Response to Reviewer 6NpQ (2/N)**
> >
> > ### **W2. The automatic graph updating may be fragile. What happens when graph structure changes unexpectedly, or computational overhead of the offline preparation phase.**
> >
> > Thank you for raising this concern.
> >
> > **Discussion of runtime stability.** In each training iteration, the computation graph is typically static (i.e., exhibits runtime stability) because the model architecture and data types remain fixed. This runtime stability of the computation graph is already leveraged by numerous optimization techniques, such as CUDA Graphs. Unexpected changes to the computation graph—such as those caused by memory corruption—can lead to failures in the entire backward pass. However, predictable graph modifications, such as dynamically selecting different branches based on input data, can be supported by our system through the pre-processing of all potential branches.
> >
> > **Offline preparation cost.** The offline preparation cost of our system is a single forward pass, for getting the overall graph structure, which is minimal compared to the entire training process.
> >
> > ### **Q1. Why does the filter operator overhead vary significantly in Table 2?**
> >
> > Thank you for the question. During training, the primary overhead of the "filter operator" arises from updating cached intermediate data using a filtering mask. Currently, this update is performed layer by layer, so the associated cost increases with model size, explaining why the 8B model in Table 2 exhibits higher overhead than smaller models. However, our experimental results indicate that the cost of filtering is substantially lower than the reduction in training time, resulting in a clear improvement in end-to-end efficiency. We appreciate the reviewer’s valuable question and plan to further reduce this overhead in the future—for instance, by overlapping the filtering process with computation using two micro-batches: while one updates the graph, the other executes the forward computation.
> >
> > ### **Q2. Why only evaluate Qwen2.5-1.5B but not Qwen2.5-7B?**
> >
> > Thank you for the question. We add Qwen2.5 7B in the experiments and it shows similar efficiency gain.
> >
> > |   Model    |     Training Method     | Forward | Filter Op |           Backward           |            Total             |
> > | :--------: | :---------------------: | :-----: | :-------: | :--------------------------: | :--------------------------: |
> > | Qwen2.5-7B | Regular Training (TP-4) | 362.54  |    NA     |            811.26            |           1173.81            |
> > | Qwen2.5-7B |       Our System        | 378.61  |   12.84   | 442.25 ($\downarrow 45.5$%) | 833.72 ($\downarrow 28.9$%) |
> >
> > ### **Summary of revision**
> >
> > We extend our sincere gratitude to the reviewer for your valuable and constructive comments, which have enabled us to significantly improve the manuscript. Based on your comments, we have made the following revisions to the paper.
> >
> > - Section 5
> >   - We add efficiency evaluation on larger models (i.e., 14B$\sim$40B).
> >   - We add results of Qwen2.5-7B to Table 2.
> >   - We analyze the overhead of the filtering operator in detail.
> > - Section 6
> >   - We discuss the future direction of combining our system with MoE
> > - Appendix D
> >   - We update the analysis of combining our system with various parallelism strategies.

---

> > > ### Comment · Reviewer_6NpQ · 2025-11-26
> > >
> > > Thanks for the response, I will keep my positive score.

---

> > > > ### Author Response · Authors · 2025-11-28
> > > > **Response to Reviewer 6NpQ**
> > > >
> > > > Thank you for your reply and continued support of our manuscript. We sincerely appreciate your time and positive consideration.

---

### Author Response · Authors · 2025-12-02
**Summary of rebuttal and discussion period**

We thank the reviewers for their valuable and insightful comments. Below, we summarize the key improvements from the discussion period:

- **Efficiency** evaluation on larger LMs (in response to Reviewers 6NpQ, P85p, VSJV). In addition to our experiments with an 8B parameter model (Llama3.1), we add experiments with larger 14B (Qwen3), 32B (Qwen3) and 40B (ALIA) parameter models. Our system scales well to larger models, consistently improving end-to-end efficiency by up to 34.7%.
- **Convergence** evaluation and analysis (in response to Reviewers VSJV, P85p, 5NLK). In addition to math reasoning domain, we add experiments in general domains. The results indicate that our system achieves a similar loss compared to existing work and does not adversely affect model convergence. We also conduct a detailed theoretical analysis of the strawman method to justify the rationality of our approach. The key insight is that the KV gradients can be compressed while the Q gradients should remain unaffected, which aligns with the rationale found in existing studies (e.g., sparse attention).
- Discussing **compatibility** with different models, frameworks, graph structures, and other optimizations (in response to Reviewers 6NpQ, P85p,5NLK). We demonstrate that our implementation adopts an approach that minimizes dependence on specific models (i.e., different graph structures) and frameworks—specifically, by directly updating nodes in the computational graph. Our implementation can support all PyTorch built-in operators and flexiably support customized operations. Our solution can seamlessly support underlying kernel optimizations, e.g., FlashAttention's tiling strategy.
- Discussing **additional cost of our system** (in Response to reviewers 6NpQ, 5NLK). We clarify the cost of offline preparation is one-time forward computation which is minimal compared to the entire training. The cost of online filtering operator is correlated with the model scale but significant smaller than the time saved in backward computation, and it could be further reduced by overlapping with the computation.
- **Clarification of algorithm novelty** and **drawbacks of the strawman approach** (in response to Reviewer 5NLK). We emphasize that the novelty of our algorithm lies not only in proposing a new memory-efficient attention compatible backward kernel that reasonably amplifies sparsity, but also in the non-trivial challenge of identifying and analyzing the problem itself. An intuitive approach would be to filter softmax activations to amplify sparsity (i.e., the strawman method); however, memory-efficient attention does not store softmax activations, and naively filtering K,Q,V activations as a substitute would unintentionally harm Q gradients. As such, these two highly challenging endeavors—identifying the core problem and its effective integration with FlashAttention—represent the principal contributions of our algorithm.
- Discussing **future work** (in response to Reviewer P85p). We discuss a variety of brilliant ideas of future work inspired by the reviewer, e.g., dynamic filtering ratio, combination with forward filtering in long-sequence training, and exploring the load-balancing issues when combining our system with MoE.

These improvements address major concerns raised by the reviewers, leading to improvements in their assessments—most notably, Reviewer VSJV stated the intention to raise score from 4 to 6, and Reviewer 6NpQ kept the positive evaluation.

---

### Meta-Review · Area_Chair_s47s · 2026-01-07

**Summary:**

This paper proposes Centrifuge, a framework (that includes algorithmic/systems co-design) for token filtering during the backward pass. Reviewers in general felt that the method made sense, tackled a practical bottleneck, and was coupled with nontrivial wallclock speedups. This can also be applied on top of existing optimized attention libraries such as FlashAttention. On the negative side, there were some concerns with regard to the initial submission only experimenting with a few models, as well as only performing experiments in the finetuning case. These were addressed experimentally during the rebuttal, and I am happy to recommend this paper to be accepted.

**Reviewer Concerns:**

Major reviewer concerns initially included:
- Limited to only a few models
- Limited to only finetuning (and not full pretraining)
- Complexity (and potential fragility) of the system.

The first two points were addressed experimentally during the rebuttal. The third point was addressed in the rebuttal text.

**Reviewer Scores:**

Reviewer VSJV stated that they would raise their score to a 6, while reviewer 6NpQ stated that they would maintain their positive score. For the other reviews, my sense is that they would have remained the same.

---

### Decision · Program_Chairs · 2026-01-26

Accept (Poster)